# Diversity and Potential Cross-Species Transmission of Rotavirus A in Wild Animals in Yunnan, China

**DOI:** 10.3390/microorganisms13010145

**Published:** 2025-01-13

**Authors:** Xiang Le, Yinzhu Tao, Binghui Wang, Yutong Hou, Yuting Ning, Junjie Hou, Ruimei Wang, Qian Li, Xueshan Xia

**Affiliations:** 1Faculty of Life Science and Technology, Kunming University of Science and Technology, Kunming 650500, China; 2Yunnan Province Key Laboratory of Public Health and Biosafety, School of Public Health, Kunming Medical University, Kunming 650500, China; 3Clinical Medical College, Dali University, Dali 671000, China

**Keywords:** Rotavirus A, interspecies transmission, mammalian rotavirus A

## Abstract

Rotavirus A (RVA) is the primary enteric pathogen of humans and many other species. However, RVA interspecies transmission remains poorly understood. In this study, we conducted a comprehensive screening and genotyping analysis of RVA in 1706 wild animal samples collected from various regions within Yunnan Province, China. A total of 24 samples, originating from wild boars, rodents, bats, and birds tested positive for RVA. Next generation sequencing and phylogenetic analyses revealed a high degree of genetic diversity and reassortment, particularly for VP4 and VP7. Strains isolated from wild boars and rodents exhibited gene segments with high similarity to those found in humans and other mammalian RVA strains, indicating that RVA may undergo interspecies transmission and reassortment, resulting in novel strains with potential risks for human infection. This study provides critical data for understanding the transmission mechanisms and the RVA host range, and highlights the pivotal role of wildlife in viral evolution and dissemination. These findings have significant implications for public health policies and emphasize the need for enhanced surveillance to prevent interspecies RVA transmission.

## 1. Introduction

Rotavirus (RV) is a double-stranded RNA virus of the Rotavirus genus within the Reoviridae family [1]. There are nine recognized Rotavirus species, designated rotaviruses A to J. Among these, rotavirus A (RVA) infects a wide range of mammals and birds, causing diarrhea and developmental issues (that is, growth retardation syndrome, leading to significant economic losses in animal husbandry) [2,3]. Rotavirus B primarily affects pigs, and has occasionally infected humans. RVA infections are particularly severe during breeding periods, increase treatment costs, reduce growth efficiency, and result in high mortality [4,5]. Pigs serve as important hosts for RVA, with infections commonly observed in intensive farming environments, where they substantially affect production efficiency [6]. Rotavirus C is associated with gastrointestinal illnesses in younger human populations, and is prevalent in animals [7]. Types D and F have been exclusively identified in avian species; type G is most commonly found in poultry [8,9]; and type H infects humans, pigs, and bats [10]. Type I has been identified in dogs, cats, and sea lions, and type J has been identified in bats [10,11,12,13].

Among the subtypes, RVA is the primary pathogen in human infections, making it a major public health concern [14]. Human RVA infections are a major cause of gastroenteritis with symptoms such as diarrhea, vomiting, and dehydration, and are a leading cause of acute gastroenteritis in infants and young children worldwide [15,16]. Notably, in developing countries, RVA is a major pathogen contributing to severe diarrhea and mortality in children under five years of age [4]. RVA is predominantly transmitted via the fecal-oral route through contact with contaminated food, water, and surfaces or through direct exposure to infected animal excreta [17].

The RVA genome comprises 11 segments, each encoding a structural protein (VP1, VP2, VP3, VP4, VP6, and VP7) or a nonstructural protein (NSP1, NSP2, NSP3, NSP4, and NSP5) [1]. VP7 and VP4 determine the G and P genotypes, respectively, and classify RVA into distinct serotypes [18]. According to the latest report of the Rotavirus Classification Working Group (RCWG) [19], to date, 42 G and 58 P genotypes have been identified [20]. In humans, the most common G genotypes are G1, G2, G3, G4, G9, and G12, whereas the predominant P genotypes are P[4], P[6], and P[8] [15]. The globally dominant G/P genotype combinations, causing nearly 90% of human RVA infections, are G1P[8], G2P[4], G3P[8], G4P[8], G9P[8], and G12P[8] [4,18,21,22]. In Asia, genotypes G1P[8], G2P[4], G3P[8], and G4P[8] have been frequently reported, although their prevalence varies by region and time [14,21,23,24]. For example, G1P[8] and G2P[4] are predominant in India, whereas G3P[8] and G4P[8] are more prevalent in Japan [23,24]. Recent studies in China have revealed an increasing trend in the prevalence of the G9P[8] genotype [21]. Research conducted in the Hubei Province between 2019 and 2020 identified the G9P[8]-E2 strain as a locally circulating strain that is closely related to similar strains in other parts of China and Japan [24,25,26].

Genetic reassortment events between porcine and human RVA strains have been reported, potentially resulting in novel strains with altered host ranges and pathogenicity. For instance, a porcine RVA G9P[19] strain identified in Guangdong Province exhibits multiple reassorted gene segments shared with human RVA [6]. Similarly, genomic and serological evidence suggests cross-species transmission of RVA between bats and humans as well as between rodents and humans [27,28,29], further underscoring the public health risks posed by interspecies transmission and reassortment.

Yunnan Province is characterized by wildlife diversity and frequent human-wildlife interactions, which increases the risk of zoonotic viral transmission. Notably, owing to potential interspecies transmission, RVA requires vigilant monitoring. Therefore, in this study, we aimed to determine the presence and genetic diversity of RVA in various wild animal species in the natural ecosystems of Yunnan. In addition, using comprehensive genomic analyses, including whole-genome sequencing and targeted gene analyses, we aimed to determine the potential risk of cross-species RVA transmission. 

## 2. Materials and Methods

### 2.1. Sample Collection 

From 2020 to 2024, 1706 samples were collected from various regions of Yun-nan Province, China. The samples were obtained from wild boars, rodents, bats, birds, and other wild mammalian species. Rodents were captured at sunset using baited live traps placed near human dwellings and retrieved before sunrise to prevent sample degradation due to non-human interference. Bats were captured using mist nets placed near their roosting sites, and the captured rodents and bats were euthanized following ethical guidelines to minimize suffering before being brought to the laboratory for dissection to collect small intestinal tissues. Wild boar samples were collected with the assistance of trained professionals by obtaining small intestinal tissues after humane culling. Fresh fecal samples were collected from birds and other wild mammals from the ground in their habitats, such as nature reserves and zoos, and immediately placed in non-inactivated viral transport medium. All collected samples were transported back to the laboratory in portable containers with dry ice and stored at −80 °C for subsequent experimental analyses. Strict protective measures were implemented throughout the sample collection process to prevent contamination and ensure the accuracy of downstream analyses.

### 2.2. Viral Nucleic Acid Extraction and Taxonomic Assignment

A suitable amount of tissue was homogenized and centrifuged to prepare a sample suspension. RNA and DNA were extracted using the Tianamp Virus RNA Kit (Tiangen, Beijing, China) and Tianamp Genomic DNA Kit (Tiangen, Beijing, China), respectively, according to the manufacturers’ instructions. The Cytochrome oxidase I (COI) gene was used to identify the animal species; detailed information is provided in Appendix A [30]. The remaining nucleic acid was stored at −80 °C for subsequent analyses.

### 2.3. RVA Detection and Next-Generation Sequencing 

RVA VP4 and VP7 genes were amplified using universal primers (detailed primer information is provided in Appendix A [31,32]. Positive samples were subjected to next-generation sequencing (NGS), performed by Magigene. The sequencing data were processed using an automated pipeline for de novo assembly, and primers were designed using Oligo 7 software to achieve complete genomic assembly. These primers were used for the validation and amplification of the RVA genome. The RVA sequences generated in this study have been submitted to NCBI GenBank (accession numbers are provided in Appendix A).

### 2.4. Sequence Alignment and Phylogenetic Tree Analysis

Reference rotavirus sequences were downloaded from NCBI GenBank (GenBank Overview, https://www.ncbi.nlm.nih.gov/genbank/, accessed on 8 October 2024). Multiple sequence alignments (MAFFT alignment and NJ/UPGMA phylogeny (https://mafft.cbrc.jp/alignment/server/, accessed on 9 October 2024) were performed using the online tool MAFFT. Molecular evolutionary analysis was conducted using MEGA-X_V10.2.6 software, which was employed to identify the best-fitting DNA/protein substitution models (Find Best DNA/Protein Models). A maximum-likelihood (ML) phylogenetic tree was constructed based on the selected model. Phylogenetic tree stability was assessed using 1000 bootstrap replicates. Sequence identity analysis was conducted using BioAdier_V1.314 to compare the nucleotide sequence identity between the isolated RVA strains and reference RVA strains. This provided insights into genetic relatedness and sequence diversity.

## 3. Results

### 3.1. RVA Identification and Host Range 

From 2020 to 2024, 1706 animal samples were collected from various regions of Yunnan Province, China. Sampled species included rodents, wild boars, bats, birds, and other mammals. Samples were considered RVA-positive only when both the VP4 and VP7 genes were simultaneously detected. The overall RVA positivity rate across all samples was 1.4% (24/1706), and the 24 RVA-positive samples were derived from four animal groups: wild boars, rodents, bats, and birds. The RVA positivity rates for each species were as follows: wild boars (7/90, 7.78%), rodents (1/254, 0.39%), bats (14/492, 2.85%); and birds (2/648, 0.31%). Species information is provided in Appendix A. Phylogenetic analyses of VP4 and VP7 were conducted to determine the genotypes of the identified RVA strains. Among wild boars, the RVA genotypes included one strain of G3[P13], one strain of G5[P13], and five strains of G3[P13]. In rodents, bats, and wild birds, G3P[10], G3P[3], and two G34P[17] strains, respectively, were identified.

### 3.2. Next-Generation Sequencing and Genotype Constellations of RVA

To further characterize RVA strains identified in wildlife from Yunnan Province, representative strains from each animal species were subjected to genomic segment sequencing. Detailed sequencing information is provided in Appendix A. These analyses enabled the determination of full genotype constellations of the identified RVA strains. Full-length genomic sequences of three nearly complete RVA strains (ZT11-20, ZT21-30, and ZT51-59) were obtained from wild boar samples. From the LH12-21 sample, 10 RVA genomic segments were successfully sequenced, with the exception of the NSP5 segment. All sequences were validated using PCR. Each segment of the identified RVA strains was compared to similar sequences in the NCBI database. Classic reference strains for each segment were selected based on the BLAST similarity results. To elucidate the evolutionary relationships, phylogenetic trees were constructed for each segment. RVA strains isolated from wild boars and rodents exhibited unique genotype constellations (Table 1). These findings suggest distinct genetic compositions and possible reassortment events among RVA strains circulating in these wildlife populations.

### 3.3. Nucleotide Sequence Identities with Closely Related RVA Strains

To further characterize the RVA strains identified in wildlife from Yunnan Province, representative strains from each animal species were subjected to genomic segment sequencing. Detailed sequencing information is provided in Appendix A. These analyses enabled the determination of full genotype constellations of the identified RVA strains, highlighting their genetic diversity and evolutionary relationships. 

For ZT11-20, the genomic segments exhibited high nucleotide sequence similarity to strains from multiple host species. The VP7 segment showed 98.16% similarity with a human-derived strain (Rotavirus long E-type, AF501578.1) identified in India, whereas VP1 showed 91.59% similarity with a bovine RVA strain (Bovine rotavirus HLJ-H3/2022/CHN, OQ807041.1) reported in China. The VP2 segment showed 95.90% similarity to an ovine strain from China, whereas VP3 showed the highest similarity (92.47%) to a giant panda strain (Giant panda rotavirus A CH-1, HQ641295.1). NSP1 and NSP2 were most similar to human RVA strains, with NSP1 sharing 97.63% similarity with Human rotavirus A R479 (GU189555.1) from China and NSP2 showing 99.53% similarity to RVA/Human-wt/THA/SKT-27/2012/G6P[14](LC055554.1) from Thailand. Other genomic segments (VP4, VP6, NSP3, NSP4, and NSP5) exhibited nucleotide similarities ranging from 95.02% to 99.04%, primarily among porcine RVA strains.

For ZT51-59, all genomic segments displayed nucleotide similarity exceeding 94.99% with their closest corresponding reference strains. The VP7 segment exhibited 96.62% similarity with a human RVA strain (RVA LL3354, 159575.1) discovered in Hebei, China, and VP3 showed 98.53% similarity with a strain (RVA NT0599, LC095937.1) reported in Vietnam. NSP3 exhibited 98.74% similarity to RVA R1954 (KF726074.1), which was identified in Wuhan, Hubei, China.

For LH12-21, among the 10 genomic segments, eight exhibited the highest nucleotide similarity to human-derived RVA strains: VP7 (93.81%), VP4 (86.51%), VP1 (90.57%), VP2 (89.69%), VP3 (90.73%), NSP1 (89.58%), NSP2 (90.89%), and NSP3 (90.80%; Table 2). Additionally, NSP4 shared 93.78% similarity with a bat-derived strain (Bat/2018/S18CXBatR24, OP963645.1), whereas VP6 shared 93.49% similarity with a feline-derived RVA strain from Japan. Notably, the host LH12-21 was identified as *Rattus tanezum* via Cyt b gene analysis. However, none of its genomic segments were similar to those of rodent-derived RVA strains, suggesting potential reassortment or cross-species transmission.

### 3.4. Phylogenetic Analysis of Rotavirus in Wild Animals

Phylogenetic trees were constructed using classical reference RVA sequences downloaded from NCBI and wildlife sample sequences obtained in this study (Figure 1). The VP7 phylogenetic tree included 24 positive sample sequences and 74 reference sequences from NCBI GenBank. The sequences clustered into two main groups: mammalian and avian RVA. The mammalian RVA VP7 genes detected in this study were classified as three genotypes—G3, G5, and G9—whereas the avian RVA sequences belonged to the G34 genotype. Within the mammalian RVA group, VP7 sequences exhibited varying phylogenetic relationships. The G3 genotype was detected in wild boars, bats, and rodents; however, sequences from different host species were relatively distant in the tree, reflecting host-specific genetic diversity. For example, the LH12-21 strain clustered closely with human RVA strains detected in China in 2013 (KU243693.1, KU243690.1), whereas ZT21-30 grouped with a human RVA strain identified in China in 2020 (PP869338.1). Bat-derived G3 strains formed two sub-clusters: one group with goat RVA (AB056650.1) and bat RVA MYAS33 (KF649188.1), and the other clustered with human RVA M2-102 (KU597744.1) and bat RVA LZHP2 (KX814942.1). The G9 genotype was represented by a single strain, ZT11-20, that clustered with a human RVA strain detected in Ho Chi Minh City, Vietnam (AB091777.1). The G5 genotype, found in wild boars, formed three distinct subclusters. For example, ZT10 was closely related to a porcine RVA strain from Hebei, China, in 2022 (PP901963.1), while ZT51-59 showed closer evolutionary proximity to a strain from Hubei, China (EF159575.1). In the avian RVA group, the VP7 sequences belonged to the G34 genotype and clustered closely with an avian RVA strain detected in New Zealand in 2021 from *Sturnus vulgaris* oral and cloacal swabs (OR645519.1). 

Phylogenetic trees were constructed using 24 positive VP4 gene sequences obtained in the present study and 82 reference sequences downloaded from NCBI GenBank. The sequences clustered into two main groups: mammalian and avian RVA. Within the mammalian RVA group, VP4 sequences were further divided into three subgroups corresponding to genotypes P[3], P[10], and P[13]. All VP4 sequences from wild boars belonged to the P[13] genotype, which was further separated into three subclusters based on their associated G genotypes. For the P[13] genotype, ZT21-30 clustered with a porcine RVA strain detected in China (OQ799714.1), while ZT11-20 grouped with a strain identified in domestic pigs in Vietnam (KX363348.1). The remaining P[13] strains were more closely related to the strain reported in 2023 from pigs in Xuzhou, China (PP580381.1). The rodent RVA strain was identified as the P[10] genotype and clustered with the RVA strains detected in humans in India (EF672556.1, JQ863312.1). Bat-derived RVA strains were classified as the P[3] genotype and formed two distinct subclusters: one group with an RVA strain found in *Rhinolophus pearsonii* in Guangxi, China, in 2013 (OR868768.1), and the other cluster with a strain identified in *Rhinolophus hipposideros* (MSLH14, KC960622.1). The avian RVA sequences belonged to the P[17] genotype and were closely related to the avian rotavirus strain PO-13 (AB009632.2) in the phylogenetic tree.

Phylogenetic analyses of VP6, VP1, VP2, VP3 (Figure 2) and NSP1–NSP5 (Figure 3) genes revealed diverse genotypes and evolutionary relationships among the RVA strains from wild boars and rodents. VP6 sequences from wild boar strains were classified as I5, forming three distinct branches. Although some strains showed close evolutionary relationships with the porcine strains from China (OR127198.1), others were more dispersed and distant within the phylogenetic tree. The rodent strain was classified as I3 and clustered with the porcine strain (KU243656.1). For VP1, the strains spanned seven genotypes, with the wild boar strains distributed across R1 and R2, and the rodent strain classified as R3. Some strains showed a close phylogenetic proximity to human strains from Russia (MT876637.1). The VP2 sequences were distributed among the C1, C2, and C11 genotypes, with the rodent strain belonging to C11, whereas the wild boar strains were primarily found in C1 and C2. VP3 gene analysis revealed that the wild boar strains were of the M1 genotype, whereas the rodent strains were classified as M11. Interestingly, the VP3 sequence of ZT11-20 shared 93.34% nucleotide similarity with a giant panda strain isolated in 2008 (HQ641295.1), although it did not cluster with the panda strain in the phylogenetic tree, indicating divergence. NSP1 sequences were classified into A1, A8, and A22 genotypes, with wild boar strains primarily belonging to A1 and A8, and the rodent strain belonging to A22. NSP2 sequences spanned the N1, N2, and N3 genotypes, with wild boar strains mostly found in N1, whereas some belonged to N2. The rodent strain was classified as N3. The NSP3 sequences from the wild boar strains were all classified as T1, whereas those from the rodent strain belonged to T14. Despite the low nucleotide similarity (68.66%) between some strains and human strains (KF726075.1), they showed relatively close evolutionary relationships. The NSP4 sequences indicated that the wild boar strains were classified as E1, showing proximity to human strains (K02032.1), whereas the rodent strain was classified as E3. NSP5 sequences from wild boar strains were classified as H1, which is closely related to an infant diarrheal strain from China (KF041439.1). 

## 4. Discussion

RVA is a major diarrhea-causing pathogen in infants and young children, often leading to severe dehydration, and in extreme cases, death [4,15]. Owing to its high pathogenicity and widespread transmissibility, RVA is a global public health threat. Understanding the transmission pathways, host ranges, and genetic variation is essential for developing effective prevention and control strategies [2,28,31]. 

In this study, 1706 samples were collected from various animal hosts in Yunnan Province, including rodents, wild boars, bats, birds, and other mammals. RVA was detected in 24 samples, with an overall positivity rate of 1.4% (24/1706). Positivity rates varied across hosts, with wild boars exhibiting the highest rate (7.78%) [6,31], which is potentially linked to their ecological behavior and habitats. Rodents showed a lower positivity rate (0.39%) than that observed in previous studies [29], suggesting geographic and ecological variability. In bats, RVA positivity rates ranged from 1.63 to 13.64% across species, which is consistent with earlier studies indicating species-specific susceptibility [3]. Wild birds showed a positivity rate of 0.31%, with infections detected only in *Passeriformes* (1.39%), which was significantly lower than previously reported rates in domestic and wild avian populations [8]. 

The identified RVA genotypes were G3P[3], G3P[10], G3P[13], G5P[13], and G9P[13] in mammals, and G34P[17] in birds. The observed host specificity of RVA susceptibility suggests that certain species, such as wild boar, serve as significant reservoirs for RVA transmission [29,31]. Phylogenetic analysis revealed complex evolutionary relationships among RVA strains, even when similar genotypes were detected across different hosts. For example, although G3 genotypes have been identified in multiple hosts, their genetic distances indicate host-specific evolutionary pathways [4,33,34]. Wild boar and rodent RVA strains exhibit high similarity to human RVA strains, suggesting potential interspecies transmission and reassortment events that pose a risk for human infection [2,17,35,36].

NGS revealed extensive genetic diversity in the RVA strains isolated from wild boars and rodents, particularly in VP4 and VP7. Phylogenetic and genomic analyses have highlighted the broad host range for RVA infection and its complex genetic structure, driven by frequent reassortment events [6,35,37]. Natural reservoirs and host diversity facilitate interspecies transmission, which is further compounded by reassortment during coinfection with multiple RVA strains. These processes led to the emergence of novel strains with enhanced transmissibility and pathogenicity [27,38].

Genomic analysis of the wild boar-derived strain ZT11-20 showed high nucleotide similarity with RVA strains from cattle, sheep, and other mammals, suggesting that it is a reassortant strain formed through multiple interspecies transmissions [6,31]. Similarly, the rodent-derived strain, LH12-21, exhibited high nucleotide similarity with human, bat, and feline RVA strains, indicating that it is a reassortment strain. Notably, none of the genomic segments of LH12-21 showed similarities to rodent-specific strains, suggesting that its genome may have originated from multiple mammalian RVA strains and subsequently adapted in rodents [17,27,29]. Studies have shown that homologous rotaviruses exhibit stronger replication efficiency and capability when infecting their homologous hosts compared to heterologous rotaviruses. Enkai Li, Ningguo Feng, and colleagues validated this result through animal experiments [39,40].

These findings provide robust evidence that the RVA strains identified in this study are products of interspecies transmission and reassortment, with potential risks of human infection [41]. This study has several limitations, including the inability to determine whether the detected RVA strains are representative of broader populations of wild boar and rodents. The role of environmental factors and coinfections in facilitating reassortment events warrants further investigation. Additionally, no human RVA sequences from the Yunnan Province closely matched the strains reported in this study, highlighting the need for continued surveillance. This study provides valuable evidence for RVA reassortment in wild animals in Yunnan Province, enhancing our understanding of RVA transmission and evolution in natural ecosystems. The unique geographical and ecological characteristics of Yunnan make it an ideal location for studying interspecific viral transmission. These findings emphasize the critical role of wildlife in the evolution and spread of RVA, and offer important implications for RVA control strategies and zoonotic disease prevention. Future research should increase the sample size, include seasonal and species-specific dynamics, and investigate the evolutionary rates and adaptive mutations in RVA gene segments. Strengthening the surveillance of RVA interspecies transmission will provide scientific support for public health policies and vaccine development, ultimately mitigating the risk of emerging infectious diseases, specifically, RVA-associated diseases [14,22].

## 5. Conclusions

Through RVA screening and genotyping in wildlife samples obtained from the Yunnan Province, this study highlights the broad host range, species-specific characteristics, and complex genotypic diversity of RVA. These findings provide new insights into the mechanisms of RVA transmission and serve as an important reference for monitoring interspecies RVA transmission in wildlife. Whole-genome sequencing and phylogenetic analyses revealed significant genetic diversity and reassortment events among RVA strains in wildlife. Notably, RVA strains from wild boars and rodents exhibited high genetic similarity to strains from other species, suggesting that RVA may cross species barriers between wildlife, livestock, and humans. This underscores the critical role of wildlife in the evolution and dissemination of RVA and emphasizes the need for enhanced surveillance to mitigate the risks associated with interspecies transmission.

## Figures and Tables

**Figure 1 microorganisms-13-00145-f001:**
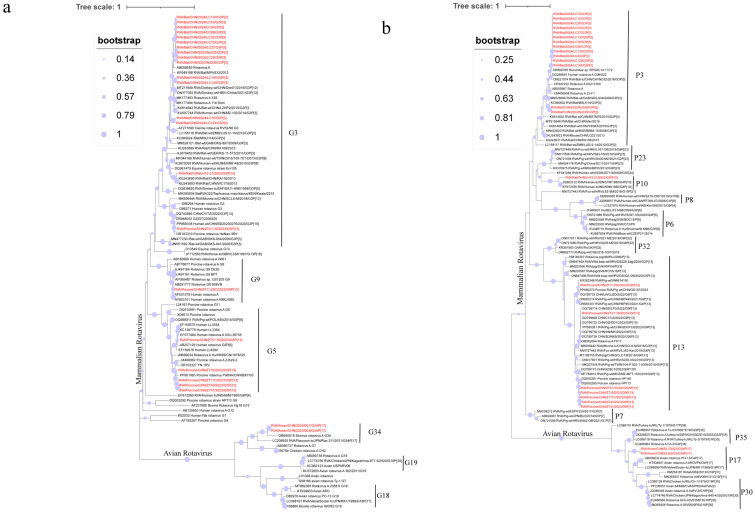
The phylogenetic relationship between the VP7 (**a**) and VP4 (**b**) and closely related strains and genotype reference strains screened in this study was investigated (red font). (**a**) Based on the VP7 gene, MEGA-X was constructed using the maximum likelihood method (ML), and model (GTR+G), and bootstrap analysis was performed 1000 times. (**b**) Based on the VP4 gene, MEGA-X was constructed using the maximum likelihood method (ML) and model (GTR+G), and bootstrap analysis was performed 1000 times.

**Figure 2 microorganisms-13-00145-f002:**
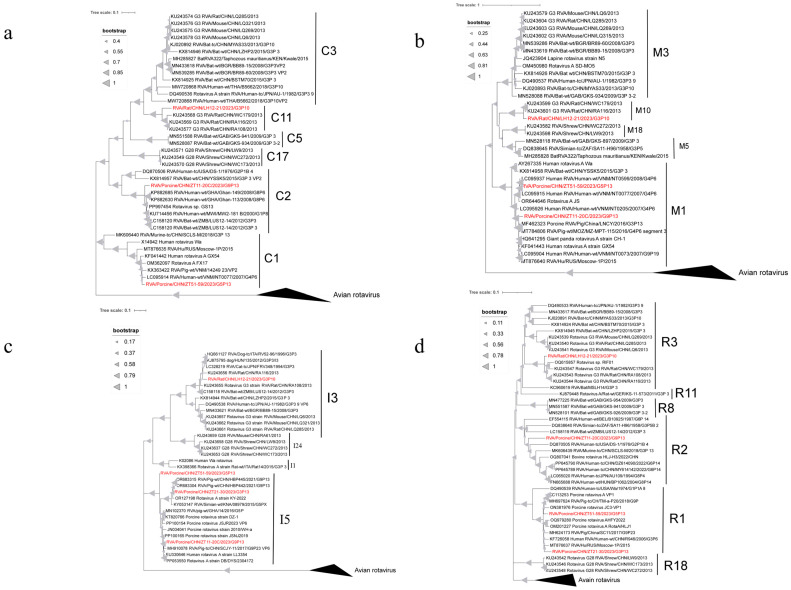
The phylogenetic relationship between the structural genes VP1 (**a**), VP2 gene (**b**), VP3 gene (**c**) and VP6 gene (**d**) of rotavirus A in wild animals screened in this study with closely related strains and genotype reference strains (red font). (**a**) Based on the VP1 gene, MEGA-X was constructed using the maximum likelihood method (ML), and model (GTR+G+I), and bootstrap analysis was performed 1000 times. (**b**) Based on the VP2 gene, MEGA-X was constructed using the maximum likelihood method (ML), and model (GTR+G), and bootstrap analysis was performed 1000 times. (**c**) Based on the VP3 gene, MEGA-X was constructed using the maximum likelihood method (ML), and model (GTR+G+I), and bootstrap analysis was performed 1000 times. (**d**) Based on the VP6 gene, MEGA-X was constructed using the maximum likelihood method (ML), and model (T92+G), and bootstrap analysis was performed 1000 times.

**Figure 3 microorganisms-13-00145-f003:**
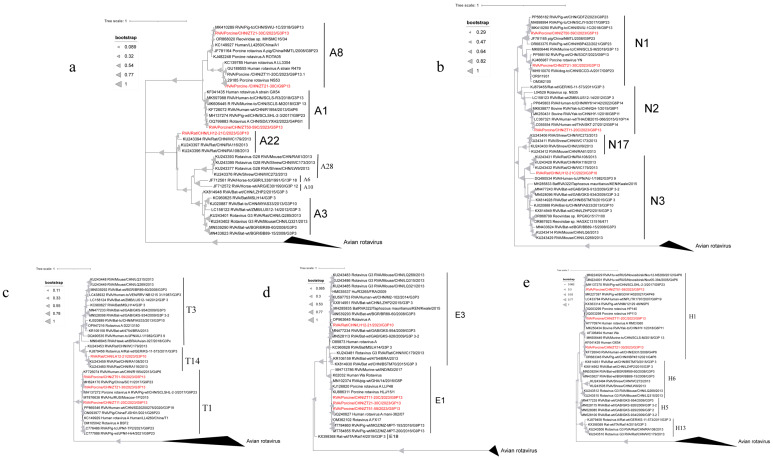
The phylogenetic relationship between the non-structural genes NSP1 (**a**), NSP2 gene (**b**), NSP3 gene (**c**), NSP4 gene (**d**) and NSP5 gene (**e**) of rotavirus A in wild animals screened in this study and closely related strains and genotype reference strains (red font). (**a**) Based on the NSP1 gene, MEGA-X was constructed using the maximum likelihood method (ML), and model (GTR+G), and bootstrap analysis was performed 1000 times. Based on the NSP2 gene, MEGA-X was constructed using the maximum likelihood method (ML), and model (TN93+G), and bootstrap analysis was performed 1000 times. (**c**) Based on the NSP3 gene, MEGA-X was constructed using the maximum likelihood method (ML), and model (GTR+G), and bootstrap analysis was performed 1000 times. (**d**) Based on the NSP4 gene, MEGA-X was constructed using the maximum likelihood method (ML) and and model (HKY+G) for 1000 bootstrap analyses. Based on the NSP5 gene, MEGA-X was constructed using the maximum likelihood method (ML), and model (T92+G), and bootstrap analysis was performed 1000 times.

**Table 1 microorganisms-13-00145-t001:** Genotype constellations of RVAs in this study.

Strains	VP7	VP4	VP6	VP1	VP2	VP3	NSP1	NSP2	NSP3	NSP4	NSP5
ZT11-20	G9	P[13]	I5	R2	C2	M1	A8	N2	T1	E1	H1
ZT21-30	G3	P[13]	I5	R1	Cx	Mx	A8	N1	T1	E1	H1
ZT51-59	G5	P[13]	I5	R1	C1	M1	A1	N1	T1	E1	H1
LH12-21	G3	P[10]	I3	R3	C11	M10	A22	N3	T14	E3	Hx

**Table 2 microorganisms-13-00145-t002:** Nucleotide Sequence Similarity Analysis of LH12-21 Genomic Segments.

			Strain exhibiting highest identity				
Strain	Gene	Genetype	Strain name	Accession no.	Nucleotide identity(%)	Host	Country
LH12-21	VP7	G3	RVA/Rat/CHN/RA116/2013G3	KU243690	93.81	*Homo sapiens*	China
	VP4	P[10]	RVA/Human-tc/IDN/57M/1980/G4P[10]	JQ863312	86.51	*Homo sapiens*	Indonesia
	VP6	I3	RVA/Cat-tc/JPN/FRV348/1994/G3P[3]	LC328219	93.49	*Felis catus*	Japan
	VP1	R3	RVA/Rat/CHN/WC179/2013	KU243547	90.57	*Homo sapiens*	China
	VP2	C11	RVA/Rat/CHN/RA116/2013	KU243569	89.69	*Homo sapiens*	China
	VP3	M10	RVA/Rat/CHN/WC179/2013	KU243599	90.73	*Homo sapiens*	China
	NSP1	A22	RVA/Rat/CHN/RA116/2013	KU243397	89.58	*Homo sapiens*	China
	NSP2	N3	RVA/Rat/CHN/WC179/2013	KU243432	90.89	*Homo sapiens*	China
	NSP3	T14	RVA/Rat/CHN/RA108/2013	KU243459	95.80	*Homo sapiens*	China
	NSP4	E3	Bat/2018/S18CXBatR24	OP963645	93.78	*Rhinolophus marshalli*	China

## Data Availability

All the relevant data are included in the manuscript in an aggregated forma.

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
