# Peer review of "Diversity and Potential Cross-Species Transmission of Rotavirus A in Wild Animals in Yunnan, China"

_microorganisms, 2025, doi:10.3390/microorganisms13010145_

Round 1

Reviewer 1 Report

Comments and Suggestions for Authors

I have only few details to change but overall it is a really good work, with a high impact for human and wildlife health.

I have only few details to change but overall it is a good job, with high impact for human and wildlife health. This research opens new doors for further and necessary researches. 

Line 40-42: why do you say G has been exclusively identified in avian species but you put between bracktes mammals and other different species? Maybe you wanted to say exclusively avian species but it has been occasionally isolated in other mammals species like pigs, cattle, horses…

Introduction: well focused, you describe the importance of the pathogen and the risk of zoonosis. Well done.

2. materials and methods

2.1 Sample collection

You should mention the nature of the samples in the first sentence. 1706 samples of what? Serum, feces, organs? You only say in rodents how you capture and euthanize them but not what kind of samples are you taking from them. How did you capture wild boar? You only mention that it was safe...did you dart them? Did you hunt them? And again, what kind of samples?

Fecal samples from other wild mammals...where? From the soil? From the nest? From the animal directly? How did you did it? Maybe you should describe it with more detail.

3. Results

3.1. RVA identification and host range

Again...you do not mention the nature of the samples. Maybe you should mention the best samples for determination of this pathogen. It can be a clue for results. Review this fact in the article.

Rest of the article: ok

Author Response

Comments 1:  Line 40-42: why do you say G has been exclusively identified in avian species but you put between bracktes mammals and other different species? Maybe you wanted to say exclusively avian species but it has been occasionally isolated in other mammals species like pigs, cattle, horses…

Response 1:  Thank you for pointing this out. We agree with this comment. Therefore, the erroneous species information has been removed in lines 40–41. And the necessary modifications have been made in the text. Types D and F have been exclusively identified in avian species, whereas type G is most commonly found in poultry.

Comments 2:  You should mention the nature of the samples in the first sentence. 1706 samples of what? Serum, feces, organs? You only say in rodents how you capture and euthanize them but not what kind of samples are you taking from them. How did you capture wild boar? You only mention that it was safe...did you dart them? Did you hunt them? And again, what kind of samples?

Fecal samples from other wild mammals...where? From the soil? From the nest? From the animal directly? How did you did it? Maybe you should describe it with more detail.

Response 2:  Thank you very much for pointing out this issue with our content. We agree with this comment. We have provided a clearer description in 2. Materials and Methods, 2.1 Sample Collection, with the following modifications: Lin 84-98: From 2020 to 2024, 1,706 samples were collected from various regions of Yun-nan Province, China. The samples were obtained from wild boars, rodents, bats, birds, and other wild mammalian species. Rodents were captured at sunset using baited live traps placed near human dwellings and retrieved before sunrise to prevent sample degradation due to non-human interference. Bats were captured using mist nets placed near their roosting sites, and the captured rodents and bats were euthanized following ethical guidelines to minimize suffering before being brought to the laboratory for dissection to collect small intestinal tissues. Wild boar samples were collected with the assistance of trained professionals by obtaining small intestinal tissues after humane culling. Fresh fecal samples were collected from birds and other wild mammals from the ground in their habitats, such as nature reserves and zoos, and immediately placed in non-inactivated viral transport medium. All collected samples were transported back to the laboratory in portable containers with dry ice and stored at -80 °C for subsequent experimental analyses. Strict protective measures were implemented throughout the sample collection process to prevent contamination and ensure the accuracy of downstream analyses.]

Comments 3:  3. Results 3.1. RVA identification and host range Again...you do not mention the nature of the samples. Maybe you should mention the best samples for determination of this pathogen. It can be a clue for results. Review this fact in the article.

Comments 3:  Thank you very much for pointing out the issues with our images. We have made the necessary adjustments and re-uploaded the images after improving their resolution (DPI). Additionally, we used high-resolution images that can be viewed clearly when zoomed in. For submission, we provided an alternative version to the editor as per the required image format. These images will not be displayed here. Once again, we sincerely appreciate your valuable feedback!

Reviewer 2 Report

Comments and Suggestions for Authors

Diversity and Potential Cross-Species Transmission of Rotavirus A in Wild Animals in Yunnan, China

In this study, Rotavirus A (RVA) was investigated as the primary enteric pathogen of humans and many other species.  The study aimed to understand RVA interspecies transmission. A comprehensive screening and genotyping analysis of RVA in 1,706 wild animal samples collected from various regions within Yunnan Province, China.

 This study provides critical data for understanding the transmission mechanisms and the RVA host range and highlights the pivotal role of wildlife in viral evolution and dissemination.

The study is well designed and well executed. The materials and method section has all the details. 

The only revision is the figures. Some figures (Fig. 1, Fig. 2, and Fig. 3) needed to be clear, very small font. 

Author Response

Comments 1:  The only revision is the figures. Some figures (Fig. 1, Fig. 2, and Fig. 3) needed to be clear, very small font. 

Response 1:  Thank you very much for pointing out the issues with our images. We have made the necessary adjustments and re-uploaded the images after improving their resolution (DPI). Additionally, we used high-resolution images that can be viewed clearly when zoomed in. For submission, we provided an alternative version to the editor as per the required image format. These images will not be displayed here. Once again, we sincerely appreciate your valuable feedback!

Reviewer 3 Report

Comments and Suggestions for Authors

The manuscript by Le et al. conducted a screening and genotyping analuysis of rotavirus in different animal species from Yunnan province and found that RV strains isolated from boars and rodents exhibited gene segments with high similarity to those found in humans. This is a good study on an important topic. However, I have some suggestions in my opinion that may improve this manuscript.

1. The authors should perform in vitro experiments to confirm RV strains isolated from animals can infected human using human intestinal organoid.

2. The resolution of Figures should be improved. 

3. The paper titled Rhesus rotavirus NSP1 mediates extra-intestinal infection and is a contributing factor for biliary obstruction is very relevant to this paper and should be discussed. 

Author Response

Comments 1:  The authors should perform in vitro experiments to confirm RV strains isolated from animals can infected human using human intestinal organoid. 

Response 1:  Thank you very much for your constructive suggestion. However, we regret to inform you that we are currently unable to perform the suggested in vitro experiments to confirm whether the RV strains isolated from animals can infect humans using human intestinal organoids due to several limitations. Our laboratory lacks the necessary equipment and facilities to culture and utilize human intestinal organoids for such experiments. The preparation, maintenance, and application of human intestinal organoids require specialized laboratory infrastructure and highly specific protocols, which we currently do not have access to. That said, we recognize the importance of the suggested experiment and agree that it would provide valuable insights into the zoonotic potential of the isolated RV strains. We are actively seeking collaborations and resources to develop this capability in the future. Once these conditions are met, we will prioritize conducting these experiments to address the question more comprehensively.

We greatly appreciate your understanding of our current limitations and your thoughtful suggestion, which has given us clear direction for the next steps in our research.

Comments 2:  The resolution of Figures should be improved. 

Response 2:  Thank you very much for pointing out the issues with our images. We have made the necessary adjustments and re-uploaded the images after improving their resolution (DPI). Additionally, we used high-resolution images that can be viewed clearly when zoomed in. For submission, we provided an alternative version to the editor as per the required image format. These images will not be displayed here. Once again, we sincerely appreciate your valuable feedback!

Comments 3:  The paper titled Rhesus rotavirus NSP1 mediates extra-intestinal infection and is a contributing factor for biliary obstruction is very relevant to this paper and should be discussed. 

Response 3:  Thank you very much for providing this paper. When writing our article, we did not fully consider this aspect. After conducting an in-depth comparison and analysis between this paper and our submitted manuscript, we have added relevant discussions in the text and addressed the shortcomings in our study with supplementary explanations and improvements. The specific revisions are as follows: Lin 327-330: Studies have shown that homologous rotaviruses exhibit stronger replication efficiency and capability when infecting their homologous hosts compared to heterologous rotaviruses. Enkai Li, Ningguo Feng, and colleagues validated this result through animal experiments.
